# Neuropsychiatric and Cognitive Deficits in Parkinson’s Disease and Their Modeling in Rodents

**DOI:** 10.3390/biomedicines9060684

**Published:** 2021-06-17

**Authors:** Mélina Decourt, Haritz Jiménez-Urbieta, Marianne Benoit-Marand, Pierre-Olivier Fernagut

**Affiliations:** Laboratoire de Neurosciences Expérimentales et Cliniques, Institut National de la Santé et de la Recherche Médicale, Université de Poitiers, INSERM U1084, 86000 Poitiers, France; melina.decourt@univ-poitiers.fr (M.D.); haritz.jimenez@univ-poitiers.fr (H.J.-U.); marianne.benoit.marand@univ-poitiers.fr (M.B.-M.)

**Keywords:** Parkinson’s disease, non-motor symptoms, rodent models

## Abstract

Parkinson’s disease (PD) is associated with a large burden of non-motor symptoms including olfactory and autonomic dysfunction, as well as neuropsychiatric (depression, anxiety, apathy) and cognitive disorders (executive dysfunctions, memory and learning impairments). Some of these non-motor symptoms may precede the onset of motor symptoms by several years, and they significantly worsen during the course of the disease. The lack of systematic improvement of these non-motor features by dopamine replacement therapy underlines their multifactorial origin, with an involvement of monoaminergic and cholinergic systems, as well as alpha-synuclein pathology in frontal and limbic cortical circuits. Here we describe mood and neuropsychiatric disorders in PD and review their occurrence in rodent models of PD. Altogether, toxin-based rodent models of PD indicate a significant but non-exclusive contribution of mesencephalic dopaminergic loss in anxiety, apathy, and depressive-like behaviors, as well as in learning and memory deficits. Gene-based models display significant deficits in learning and memory, as well as executive functions, highlighting the contribution of alpha-synuclein pathology to these non-motor deficits. Collectively, neuropsychiatric and cognitive deficits are recapitulated to some extent in rodent models, providing partial but nevertheless useful options to understand the pathophysiology of non-motor symptoms and develop therapeutic options for these debilitating symptoms of PD.

## 1. Introduction

Parkinson’s disease (PD), one of the most frequent neurodegenerative diseases, is not only characterized by motor symptoms including akinesia, bradykinesia, tremor, rigidity, and postural instability, but also by several debilitating non-motor symptoms (NMS) that may precede motor dysfunctions by several years. This prodromal phase may last 20 years or more [1] depending on clinical considerations. During this period and the development of PD, various NMS, often misdiagnosed, occur such as olfactory dysfunction (hyposmia, anosmia), gastro-intestinal disorders (constipation), sleep disorders, neuropsychiatric disorders (depression, anxiety and apathy) and cognitive dysfunction (executive dysfunctions, memory and learning impairments) [2,3]. These NMS arise in up to 90% of patients and cause a significant loss of quality of life [4]. The causes of NMS are multifactorial and poorly known. They are not restricted to the degeneration of dopaminergic neurons but are also related to alpha-synuclein aggregation and propagation, and to pathological changes affecting the olfactory bulb, locus coeruleus, raphe nuclei, amygdala, basal nucleus of Meynert (NBM) or limbic and cortical structures [5,6].

In this review, neuropsychiatric and cognitive dysfunctions are covered. A comprehensive review of other NMS such as sleep disturbances, gastro-intestinal disorders and olfactory dysfunction can be found in the review by Chesselet et al. in the same issue of Biomedicines.

## 2. Overview of Neuropsychiatric and Cognitive Deficits in PD

### 2.1. Neuropsychiatric Disorders

#### 2.1.1. Depression

Depression is a common neuropsychiatric disorder of PD. The diagnosis is based on standard criteria [7], reported in the Diagnostic and Statistical Manual of Mental Disorders (DSM V). These criteria include depressed mood, decreased feelings of pleasure, loss or gain in appetite, insomnia or hypersomnia, psychomotor agitation or retardation, loss of energy, excessive or inappropriate guilt, decreased ability to think or concentrate and recurrent thoughts of death [8]. Different types and severity of depression disorders are seen in PD patients such as minor depression (patients presenting at least two of the nine criteria), major depression (patients presenting five to nine criteria) or persistent depressive disorder (major depressive disorder lasting for at least two years). Moreover, depression can precede motor symptoms [9] appearing more commonly in PD patients five years before the onset of motor symptoms than in age-matched controls [10]. Up to 70% of patients could develop depression, depending on the criteria used [7,11] (Figure 1).

Depression may appear especially because of a dysfunction in the noradrenergic and serotoninergic innervation of locus coeruleus and raphe nuclei early in the disease process. These regions are affected during stage 2 of the Braak staging that defines the progression of alpha-synuclein pathology throughout the course of the disease [5,12]. Furthermore, loss of dopaminergic neurons in the ventral tegmental area can be associated with depression in PD patients [13]. Other hypotheses involve stress hormones, inflammation and neurotrophic factors in the development of depression [14,15].

A large part of depressive PD patients are not recognized and/or treated. The benefits of selective serotonin reuptake inhibitors (SSRI) such as sertraline, citalopram or paroxetine in depressive PD patients are not always optimum [16]. Moreover, non-ergot dopamine agonist like pramipexole (PPX), rotigotine and ropinirole seem to have an antidepressant effect unlike the ergot-derived pergolide [17,18]. In the DoPaMiP study, a French cross-sectional study, the anti-depressant effect of monoamine oxidase B inhibitor (IMAO-B) was evidenced [19].

#### 2.1.2. Anxiety

In a majority of patients, anxiety and depression coexist [20]. Anxiety disorders are characterized mainly by panic attacks, social phobias (such as agoraphobia) or general anxiety disorder. Anxiety is one of the main causes of insomnia in PD patients. Anxiety significantly affects the daily activities of PD patients and is often considered as disabling as the motor symptoms. The clinical diagnosis is based on DSM V criteria [8], specific for each anxiety-related disorder, and completed by the Mini-Mental State Exam (MMSE). Few studies have used the Crown-Crisp Index, a specific test for phobic anxiety [21]. Some studies have focused on the link between anxiety and the subsequent risk of developing PD. Recent results suggest a greater prevalence of PD in the anxious population, independent of depression [10,21]. Several studies have shown that after diagnosis, 29% to 50% of PD patients present anxiety disorder and 20% of them present at least two anxiety diagnoses [10,22]. Like depression, anxiety disorders may manifest before the onset of motor symptoms [10] (Figure 1). Anxiety is more frequent in women, in patients with younger age and a modest positive correlation with PD severity can be observed (longer disease duration, motor complications) [7].

Like depressive symptoms, anxiety behaviors occur concomitantly with Lewy bodies (LB) deposition in noradrenergic and serotoninergic neurons, that are affected in the early stages of PD [5]. A recent review highlights that functional neuroimaging studies reveal that anxiety is associated with changes in limbic cortico-striato-thalomocortical circuits that are significantly affected by the disease process, accounting for a high prevalence of anxiety in PD patients [23].

Several studies have investigated the effects of pharmacological treatments of PD on anxiety-related disorders. A prospective multicenter study demonstrated that anxiety is improved after 6 months of ropinirole treatment in PD patients with motor fluctuations or dyskinesias, probably by the stimulation of D3 receptors on the mesolimbic pathway [24]. SSRI may improve anxiety disorders [25]. In contrast, the DoPaMiP study did not show a correlation between intake of any antiparkinsonian medication (Levodopa (L-DOPA), dopaminergic agonists or IMAO-B) and the presence of anxiety-related disorders [19]. In fact, depression-related anxiety can be improved by dopaminergic treatments although anxiety can remain an underlying disorder, independent of dopaminergic state.

#### 2.1.3. Apathy

Since several years, apathy has been defined as a distinct NMS of PD, that can occur independently of depression [26]. Apathy disorder can be estimated with the MMSE and the Dimensional Apathy Scale (DAS). These tools allow one to assess three different types of apathy: executive apathy (a lack of motivation for organization, attention and planification), emotional apathy (a lack of emotional motivation, indifference or emotional neutrality), and initiation apathy (a lack of initiation of thoughts or behaviors) [27]. PD patients are significantly impaired in executive and initiation apathy compared with age-matched controls [28]. Although apathy is a frequent NMS of PD, the prevalence and pathological basis are not really known and debated. At the early stages of PD, apathy occurs in 20% of patients [4] increasing to 40% after a few years of disease progression [29] and up to 60% when patients become demented [30] (Figure 1). PD patients without dementia and with apathy present a greater motor symptomatology than PD patients without apathy symptoms [31]. Moreover, this study highlighted that apathy in PD is correlated with lower education, executive dysfunctions, and depression. Taking into account the different types of apathy, several brain structures can be involved in the lack of motivation and emotions such as the prefrontal cortex, the antero-cingular cortex, the amygdala, the ventral striatum and the hypothalamus [32,33]. Clinical data suggest an improvement of apathy by the dopaminergic stimulation of D2/D3 receptors [34]. Positive effects of pramipexole and rotigotine are shown in the motivational items but using non-specific scales for apathy [18,35]. Moreover, for patients with L-DOPA treatment, apathy is reversed in the on-state compared with the off-state [36].

### 2.2. Cognitive Dysfunction

Deficits in working memory, cognitive flexibility, learning, planning, inhibitory control and attentional abilities are often described at early stages of PD, similar to patients with frontal lobe injury [37]. These symptoms are associated with alterations of dopaminergic and cholinergic systems [38]. Moreover, cognitive performances like working memory or flexibility are highly dependent on dopaminergic circuits and dopamine levels in the prefrontal area [39]. Clinical studies based on functional imaging revealed that attention and executive dysfunctions in PD are linked with changes in frontostriatal circuits [40]. Around 30% of PD patients present mild cognitive impairment at the moment of diagnosis, increasing with the progression of the disease [41]. 30% of PD patients also display social cognition impairments early in the course of the disease, as characterized by an impaired ability to perceive other people’s emotions, as well as impaired theory of mind, as characterized by the inability to infer the mental states, beliefs or desires of other people [42,43].

#### 2.2.1. Working Memory and Attention

Short-term memory refers to the information processed by the individual in a short period of time and includes working memory, which is the ability to manipulate or work with an information stored briefly. Importantly, working memory is closely related to attention and are thus considered together as a specific cognitive domain in neuropsychology [44].

Attention deficit is a common feature observed in PD patients from the very early stage of the disease, as measured by tests that are sensitive to dysfunction in executive control [45] or tasks that measure orienting of auditory and visual cues [46].

Notably, dopamine signaling in the prefrontal cortex influences executive function [47] through strong functional connections with the striatum (STR) via parallel dopamine-dependent cortico-striatal loops [48]. Thus, first subtle attentional performance deficits in PD patients without any clinically relevant cognitive impairment could be caused by altered dopaminergic tone within associative circuits innervating the prefrontal cortex [49]. Similarly, visuospatial working memory impairments can be found in recently diagnosed and untreated PD patients without any other remarkable cognitive impairment, which can be attributable to a disturbance of strategic processes and decreased attentional performance caused by striatal dopaminergic depletion and subsequent striato-frontal dysfunction [50]. Indeed, executive function/working memory and global cognition disturbances can even precede the emergence of motor signs in subjects at risk of developing PD later in age [51]. Importantly, executive dysfunction seems to be improved by treatment with dopaminergic agents [52]. However, other aspects of attention control such as reversal-learning seem to be impaired with the treatment, which points out the importance of a proper dopaminergic tone in the prefrontal cortex for adequate behaviorbehavioral control [47].

Voluntary attentional shift control depends upon the so-called “top-down” signals arising from fronto-parietal network comprising prefrontal as well as posterior parietal cortices [53]. Functional imaging studies showed that there is an activation of the fronto-parietal network in non-demented Parkinson’s patients when they perform attentional shifting tasks, but this activation seems weaker than in control subjects probably due to a reduced connectivity within prefrontal cortical regions [54]. Interestingly, a study co-registering MRI and FDG-PET scans found that cognitive decline in PD is closely correlated with a progressive pattern of overall dysfunction in frontal and parietal cortices [55]. In PD patients with mild cognitive impairment (MCI), there was also a substantial hypometabolism in regions of that later become atrophied in PD patients with dementia [55]. Although the mechanisms leading to the dysfunction of different cortical areas of PD patients are probably diverse, the progression of Lewy pathology from the brainstem to upper structures following advanced Braak stages [56] could play a key role.

On the other hand, “bottom-up” automatic orienting of attention seems to be mediated by cholinergic projections from the NBM to the cortex. Thus, the activation of this nucleus increases the attentional significance of the environmental stimuli and facilitates their detection by posterior regions of the fronto-parietal network [57]. Besides, NBM activity is in turn partly modulated by direct inputs coming from the prefrontal cortex, so it can be inferred that NBM is an essential control center for a proper bottom-up as well as top-down attentional performance [57,58].

Importantly, the NBM undergoes progressive degeneration in PD, from a modest 30% of cell loss observed in non-demented PD patients up to 70% degeneration observed in patients with dementia [58]. Therefore, this progressive degeneration directly influences the overall cortical cholinergic innervation in PD, inducing an attenuation of cortical signal processing [59]. In this regard, studies using the acetylcholinesterase inhibitor, Rivastigmine, have shown an improvement of orienting of attention, vigilance and cognitive fluctuation, with patients showing greater attentional deficits benefiting most from the treatment [60]. In addition, other neurotransmitter systems that suffer partial degeneration throughout the disease could also be implicated in the attentional dysfunction observed in PD patients, such as the ascending noradrenergic network arising from the Locus coeruleus [61].

#### 2.2.2. Long-Term Memory and Learning

Memory is a behaviorbehavioral construct that invokes all the cognitive processes implicated in the encoding, storage and retrieval of information, and depends on several systems served by different brain structures. Overall, memory can be classified into two main subtypes, short-term memory (including working memory, as discussed above) and long-term memory. Importantly, a proper processing of memory depends upon the ability of a person to orient attention to a stimulus (to allow encoding), and upon the use of executive function to allow the retrieval of a stored information in a particular context. Thus, multiple behaviorbehavioral deficits, such as the attentional deficit itself, drive the memory impairment typically associated with PD [62].

In PD, memory impairment usually refers to long-term memory impairment. Thus, long-term memory includes declarative or conscious memory and non-declarative/unconscious memories. The first is formed by episodic memory (register for self-notions that occur during life such as facts, moments or concepts) and emotional memory (register for emotionally tagged information) [63]. Non-declarative memory, in contrast, includes implicit memory abilities such as procedural learning and thus, motor skill and habit formation [64]. Importantly, these subtypes of memories are driven by different brain regions and are thus differently affected during the progression of the disease: Declarative memory is mainly driven by amygdala and the temporal lobe of the brain, while the non-declarative memory is mostly processed by frontal cortical areas, basal ganglia and cerebellum [63,64].

##### Declarative Episodic Memory

Memory seems to be the most commonly impaired cognitive domain at baseline in PD [65]. However, memory deficits in PD patients without dementia are substantially improved after cueing [66]. Thus, this may indicate that these subjects have intact ability to store the information in long term memory, but that there is an impairment in the process of accessing to this stored information. This failure of evocation rather than failure of storage is known as “retrieval failure hypothesis”. However, one study specifically assessing learning abilities in non-demented PD subjects has demonstrated that these subjects indeed display impaired ability to acquire new information [67]. Therefore, in addition to a role for executive dysfunction and decreased attentional performance [62], declarative memory dysfunction reported in the first stages of the disease could be driven by difficulties in the processing of explicit learning. In this regard, the lack of controlling for initial learning in the majority of studies addressing long-term memory performance in non-demented PD patients could directly affect the observed outcomes. This fact should be further addressed in future studies.

However, cross-sectional studies and meta-analyses have demonstrated that recognition memory is affected even with the use of cues in PD patients with dementia [68] suggesting specific long-term storage deficits in PD in at least later stages. In this context, hypoactivation of different regions of the temporal lobe can be detected in PD patients in the first stages, even in an absence of any clinically relevant mnemonic deficits [69]. Moreover, hippocampal CA1-stratum pyramidale (SP) subfield thickness predicts declarative memory impairment in PD after controlling for confounding clinical measures [70], although other studies point towards a main contribution of subfields CA2-3 and subiculum of the hippocampus for episodic recollection [71]. The lack of consensus regarding the respective contribution of CA1 vs CA2-3 on episodic recollection deficits impedes one from extracting better conclusions, thus these outcomes warrant further investigation. Similarly, a meta-analysis of voxel-based morphometry studies have demonstrated that PD patients with dementia display decreased grey-matter volume in both the medial temporal lobe and basal ganglia, and that severity of dementia correlates with left medial temporal lobe atrophy [72]. Thus, despite subtle differences observed between the studies, specific dysfunction in the different regions encoding long-term declarative memory seems to play an important role in the deficits observed, particularly in advanced PD.

Regarding neurotransmitter systems, cholinergic projections arising from the NBM seem to be important in memory encoding, and therefore PD-associated degeneration of the NBM could directly impact not only the orienting of attention to an environmental stimulus but also the cortical connections necessary for a proper encoding of these stimulus into memory (for review see [58]). However, the effect of anticholinergic drugs such as rivastigmine improving some memory outcomes should probably be interpreted as an overall improvement of the general cognitive performance rather than as a specific memory-enhancing effect [73].

##### Declarative Emotional Memory and Reward-Learning

Reward-mediated or emotional learning is a process subserved by different brain regions of the limbic system. Reward-learning is a learning process directed by the association of rewards to previously neutral stimuli, making the decision-making process to evolve from goal-directed behavior (dependent almost exclusively on rewarded stimuli) to habit-directed behavior (more dependent on emotional experience) [74]. Importantly, the whole process is directed by cortico-basal ganglia networks and spiraling midbrain-STR-midbrain projections that allow the information to be propagated in a hierarchical manner within the STR (i.e., from “reward” related behavior encoded within the nucleus accumbens to “habit” mediated behavior encoded within the dorsal STR) [75]. In addition, the whole process is influenced by dopamine as well as cortical projections that exert a cognitive influence over this adaptive decision-making [76].

In PD patients, reward-mediated learning has been observed to be similar to controls [77,78]. However, a study specifically designed to analyze if the effect of reward on learning is distinct from the effect of corrective feedback (practice) has shown that PD patients displayed significantly less reward-related learning improvements compared to healthy controls [79], suggesting that reward processing is actually altered in PD. Since these results come from medicated PD patients, and given the pivotal role of dopamine in reward-learning, it is difficult to determine whether these outcomes are associated with the disease itself or are influenced by the treatment. In this regard, a recent study in PD patients without clinical cognitive impairment has shown that reward-learning under dopaminergic treatment is different upon the motor phenotype of PD patients. Thus, in non-tremor-dominant PD, dopaminergic medication improved reward-based choice. In contrast, tremor-dominant PD patients appear to learn from punishment rather than making reward-based choices [80]. Importantly, since non-tremor dominant PD patients display dysfunctions in several cognitive domains at baseline compared with tremor-dominant PD patients [81]. Differences in reward-learning between these two cohorts of PD patients can be directly associated with overall cognitive performance, a hypothesis that should be addressed in future studies.

##### Non-Declarative Memory

It has been reported that long-term non-declarative memory is less affected in PD than declarative (episodic) memory [82] and that some aspects of implicit cognitive skill learning seems to be preserved in PD patients even in a context of apparent executive dysfunction or declarative memory impairment [83]. However, declarative memory has been much more extensively studied in PD in comparison to non-declarative memory, and studies specifically addressing non-declarative memory in PD have usually focused just on the learning process. Notably, PD patients, in contrast to control subjects, do not display a reduction in reaction/response time over sequential learning tasks [84]. Thus, striatal dopaminergic denervation, and subsequent dysfunction of cortico-basal ganglia pathways could be the main driving factor, as striatal denervation correlates with impaired procedural-learning [85] and dopamine replacement therapy improves motor learning in PD patients [86]. Particularly, the acquisition of a new motor sequence and its stabilization seems to be dependent of a proper cortico-basal ganglia pathways controlling the function of the motor cortex, which could be altered in PD. Supporting this idea, transcranial direct current stimulation of the motor cortex reverses aberrant oscillatory activity in this region, subsequently improving the dysfunctional motor learning [87].

### 2.3. Inhibitory Control

Inhibitory control is the cognitive operation involved in the ability to stop a mental process, with or without intention. This stopping can be understood as mental suppression of competing information because of limited cognitive capacity and includes both behavioral inhibition and cognitive inhibition, which have to be adequately handled in order to avoid the emergence of impulsive-compulsive behaviors [88]. Impulsivity can be defined as the tendency to act prematurely without foresight, and includes impulsive action or motor impulsivity (the inability to withhold or to stop a movement) and decision impulsivity, encompassing impulsive-choice (the inability to inhibit or defer affectively charged actions) and reflection impulsivity (the tendency to make a decision before gathering and evaluating enough information) [89].

Importantly, the mid to long-term use of dopaminergic drugs has been widely associated to the emergence of aberrant impulsive-compulsive behaviors in PD, including classic impulse control disorders (ICD), perseverative and repetitive behavioral disturbances such as punding or hobbyism as well as excessive dopaminergic drug intake or addiction, known as Dopamine dysregulation syndrome [90]. However, it remains to be elucidated which components of the behavioral constructs of impulsivity and/or compulsivity are most altered in patients with these disturbances. Besides, given the fact that dopaminergic agents are mandatory to control the motor signs of the disease, the study of impulsive-compulsive traits in untreated PD patients have been scarce. Therefore, it is not clear if pathological changes associated with PD could act as a risk factor for treatment-induced impulsive-compulsive behaviors in PD patients.

With regard to impulsive action, the lack of proper motor inhibitory control is a common finding in PD patients, presented as deficits on tasks that test cognitive control of behavioral responses such as Stop Signal Task (SST), Stop-Signal Reaction Task (SSRT), Go-NoGo task, Anti-saccade task and stroop tasks. These deficits are present both in ON and OFF dopaminergic medication. Interestingly, patients with ICD seem to be less susceptible to making impulsive responses than patients without ICD in motor action tasks, suggesting that addictive behaviors in this subset of patients would not be related to increased impulsive action [91].

On the other hand, dopaminergic treatment seems to partly improve PD-associated response inhibition deficits when on medication in those subjects with shorter disease duration (see [92] for review), suggesting that the dopamine deficiency should underlie the response inhibition deficits in PD. However, other neurotransmitters such as noradrenaline do probably play an important role. For instance, atomoxetine, a noradrenaline reuptake inhibitor, improves response inhibition in subjects with PD, apparently by increasing connectivity of prefrontal circuits [93].

Regarding impulsive choice, untreated PD patients seem to have steeper delay discounting than healthy controls [94], an outcome that has been also observed in treated PD patients without clinically relevant ICD [95]. These results suggest that dopaminergic depletion by itself could cause an increment of this impulsive trait. In contrast, a more recent study did not find differences between PD patients and controls in temporal discounting, but reported that PD patients gathered significantly less information and made more irrational choices than matched controls in the beads task, suggestive of increased reflection impulsivity [96].

### 2.4. Behavioral Flexibility

Behavioral flexibility is defined as the capacity to adjust goal-directed behavior in response to changes in the environment [97]. The success of this process of adaptation is linked to others cognitive abilities, including decision-making, response selection or inhibition, working memory and attention. Different tests have been developed to study behavioral flexibility (reversal learning, set-shifting task) with the Wisconsin Card Sorting Test (WCST) or Stroop Test [98]. The behavioral flexibility abilities recruiting several neurobiological circuits, with the prefrontal cortex (PFC), STR and frontostriatal loops being mostly involved [99,100]. PD patients and cognitively impaired patients exhibit similar impairment in their performance compared to healthy controls [101]. Moreover, dopamine is known to be an important modulator in frontostriatal circuits and plays a crucial role in behavioral flexibility [99,102]. In the context of PD, changes of PFC activity have been demonstrated with or without striatal coactivation [100]. The study of behavioral flexibility in PD patients is a difficult challenge because the assessment of basal performances is very strenuous and one of the only ways to assess the performances without medication is to perform tests during the off-period. A cognitive flexibility disorder results mainly in perseverance in previously used strategies when an environmental change has been made and they are no longer effective and appropriate. In WCST, PD patients during off-period of medication display a higher error rate and more perseverative errors than healthy matched-control [103]. The results between the off-period and the on-period of medication did not differ significantly. However, Cools and colleagues have shown that L-DOPA improves inflexibility in PD patients but increases impulsivity [104]. Another study indicates that during a task requiring high flexibility performances, only PD patients with low flexibility performances improve under dopaminergic medication [102]. These results suggest that PD and medication have a differential impact on behavioral flexibility performances.

### 2.5. Dementia

Dementia is defined as a complex NMS, occurring when the neuropsychological profile of patients is impaired in several cognitive domains. The Movement Disorders Society based the diagnosis of dementia on: impairment in attention, executive, memory and visuospatial functions [105]. In particular, the alteration of memory is necessary for the diagnosis, and is linked with cognitive disturbances such as aphasia, apraxia, agnosia or disruption of executive functions [8]. Moreover, visual hallucinations are common in PD dementia (PDD).

This NMS evolves with the progression of the disease with a prevalence of 15–40% after 5 years, 50–70% after 10 years [106,107] (Figure 1) and impairing up to 83% of patients after 20 years [108]. The main risk factors for dementia are older age and more severe parkinsonism [109]. In addition, dementia is considered to be a worsening of mild cognitive impairment.

The pathophysiological basis for the dementia in PD patients is poorly understood but subcortical lesions may be underpinning dementia at later Braak stages [5]. Links between dementia and disruptions in dopaminergic, serotoninergic and cholinergic circuits have not been fully established although these neurotransmitters play a role in cognitive functions particularly with the involvement of the caudate nucleus, the thalamus and the prefrontal cortex [110].

Acetylcholinesterase inhibitors like rivastigmine, donepezil or galantamine, first developed for Alzheimer’s disease, have modest but positive effects on cognition in PD [111,112]. Cholinergic projections from the NBM to cortical areas are affected in PD and acetylcholinesterase inhibitors resulting in an increased half-life of the remaining acetylcholine. Moreover, memantine, an N-methyl D-aspartate (NMDA) receptor antagonist, is well tolerated by PD patients and provides a modest improvement of cognitive deficits [113].

## 3. Overview of Neuropsychiatric and Cognitive Deficits in Rodent Models of PD

### 3.1. Rodent Models of Neuropsychiatric Disorders

#### 3.1.1. Depression and Anxiety

Several studies have reported depression and anxiety-like behaviors in animal models of PD. Different tests have been used to assess depressive and anxiety-like behavior in rodents, including the forced swim test, tail suspension test, open-field test, light/dark transition task, and elevated plus maze (EPM).

These two neuropsychiatric disorders were also assessed in toxin-based models generated with 6-hydroxy dopamine (6-OHDA), 1-methyl-4-phenyl-1,2,3,6-tetrahydropyridine (MPTP), and paraquat. Indeed, after bilateral infusions of 6-OHDA into the dorsal STR or into the SNc, or after 28 days subcutaneous exposure to paraquat at a dose of 0.7 mg/day, EPM and forced swim test demonstrated depressive and anxiety-like behaviors [114,115]. Unilateral 6-OHDA injection in the medial forebrain bundle (MFB) led to anxiety-like behavior with a 96% loss of TH in the STR and 87% loss of neurons in the SNc [116]. In addition, increased immobility time in the tail suspension task was reported in this model [117]. This anxiogenic phenotype was reversed by 1.5 mg/kg of diazepam, mainly in the low anxiety hemiparkinsonian rats [116] and by 14 days treatment with etazolate at a dose of 1 mg/kg [117]. In the same model, the anxiety-like behavior was also present in the open-field test and EPM [118], and they were alleviated by hesperidine, which also reduced TH positive neurons loss [119]. In the context of partial lesion of the SNc by bilateral injection of 6-OHDA (loss of 48% of TH positive cells), no effects of L-DOPA were observed on the anxiety-like behavior induced by the lesion [120]. In addition, other results suggest that chronic L-DOPA treatment can promote an anxiogenic and depressive phenotype by increasing dopamine levels at the expense of serotoninergic levels leading to imbalance in the prefrontal cortex, the amygdala and the hippocampus [121]. In contrast, when the effects of dopaminergic agonists (SKF-38393, selective agonist D1; sumanirole, selective agonist D2R; PD-128907, preferring agonist D3R and PPX, a D3/D2 agonist) were tested, the results provided evidence that these drugs can reverse the anxiety-like behavior in 6-OHDA models, with a higher implication of D3 than D2 or D1 receptor activation [122,123,124,125].

Anxiety and depression-related behaviors are also found in mice receiving intraperitoneal MPTP or intracerebral MPP+. Accordingly, MPTP at a dose of 25 mg/kg twice a week for five weeks led to anxiogenic and depressive behaviors, as assessed by the open-field test and the tail suspension task, together with alpha-synuclein accumulation in many brain areas [126]. The same results were found with the same behavioral tests in the MPP+ model by infusion at a dose of 1.8 µg, and were reversed by pre-treatment with agmatine at doses of 0.0001, 0.01 and 1 mg/kg [127,128]. However, 4 intraperitoneal injections of MPTP at a dose of 20 mg/kg (2 h interval) led to an increase of immobility in the tail suspension task but a lack of anxious behavior was reported in the light/dark transition task 30 days following MPTP injections [129].

It has been demonstrated that VMAT-2 mice-deficient model, with a progressive decrease of striatal dopamine levels associated with a decrease of dopaminergic neurons in the SNc and alpha-synuclein aggregation, also show anxiety-like and depressive age-dependent behaviors [130]. Time spent in EPM open arms was lower in VMAT-2 deficient-mice than wild type (WT) animals at the age of 6 months whereas mice did not display difference at the age of 12–15 months. At the age of 6 months, no difference was observed in VMAT-2 deficient mice and WT group in forced swim test and tail suspension test [130,131]. However, immobility times were higher in VMAT-2 deficient-mice than WT mice at the age of 12–15 months. In addition, desipramine, a tricyclic antidepressant, had positive effects on immobility times at a dose of 20 mg/kg in both genotypes and in VMAT-2 deficient mice at an acute dose of 5 mg/kg. A second study [132] tested mice at the age of 3–5 months and found no anxiety-like behaviors after open-field test and light/dark transition task. In contrast, VMAT-2 deficient mice presented an increase immobility in the forced swim and tail suspension tests, decreasing after imipramine injection at doses of 5 and 10 mg/kg and after 30 mg/kg of fluoxetine.

In the genetic model of parkin deficient mice, induced by deletion of exon 3, these mice did not show significant motor impairment in the rotarod test and open-field locomotor activity at the age of 6, 12, 15, 18 and 21 months [133]. Nevertheless, in light/dark transition task and in open-field with thigmotaxis behavior, parkin deficient mice displayed anxiety-like symptoms at 6 and 15 months. However, other studies did not report an anxiogenic profile in parkin deficient mice with the same behavioral tests but confirmed the absence of locomotor alterations [134,135].

Neuropsychiatric disorders develop in transgenic mice carrying hA53T mutation by the mouse prion promoter. Open-field testing demonstrated more time spent in the periphery of the open-field than heterozygous A53T mice and controls at the age of 3 and 6 months [136], allowing for the conclusion that this model displays anxiety-like behavior correlating with motor impairments and diminution of dopamine transporter (DAT) level in the STR. Moreover, during the tail suspension task, homozygous A53T mice spent more time in immobility than the other groups at the age of 6 months. In contrast, another study showed a reduced anxiety-like behavior in this model at the age of 12 months with the open-field test revealing an increase of time spent in the central area [137]. Several studies demonstrated similar results at the age of 12 months with open-field test and EPM [138,139]. Finally, old-age A53T transgenic mice show hypoanxiety-like behaviors albeit increased anxiety occurs at 2 months in the same animals. This phenomenon remains unexplained but may be due to age-related changes in DAT expression and serotonin transporter (SERT) trafficking. To end with genetic models of PD, BAC transgenic rats in which human wild-type α-synuclein was overexpressed display age-dependent accumulation of full-length and C-terminal truncated α-synuclein aggregates into insoluble fibers in the brain (particularly in the STR), along with early disturbances in novelty-seeking (indicative of increased anxiety) [140].

Viral-mediated strategies have also been used for modelling neuropsychiatric disorders in PD. Indeed, an injection of 3 µL of Adenoassociated viral vector (AAV)-6 expressing alpha-synuclein in the SNc led to forced swim test impairment from 3 weeks after surgery but not in EPM, associated with motor impairments [141]. The absence of hyperanxious-like behavior in the EPM behavioral test is also suggested by other studies using the AAV-2 alpha-synuclein model [114,142]. Moreover, a lack of depressive state in this model can be observed in the forced swim test, unlike in a previous study [114].

#### 3.1.2. Apathy

In experimental models, apathetic behavior (or anhedonia) can be assessed with the sucrose preference test (SPT) with different protocols adapted from Willner’s protocol [143] and by operant sucrose administration [124]. In SPT, some studies performed food and water deprivation before testing and the sucrose concentration is not always the same, which can affect the motivational process. To avoid a place preference in SPT, the two bottles (water and sucrose) are switched during trials. In the operant procedure, a 2% sucrose solution is proposed to rats in free-access during one week and then during 7 days of self-administration following by 2 days of progressive ratio task. With this protocol the motivational process is taken into account.

Regarding 6-OHDA models, bilateral infusions in the SNc did not induce an apathetic-like behavior in a SPT with 3% sucrose solution during 1 h exposure [114]. However, several studies demonstrated that bilateral infusions of 6-OHDA in the SNc led to a decreased sucrose preference, with two different concentrations (1% or 0.5%) [144,145]. In addition, bilateral infusions of 6-OHDA in the dorsal STR also led to a decrease of sucrose (0.5% and 0.8%) consumption [115], reversed by the small conductance calcium-activated K+ blocker apamin at a dose of 0.1 or 0.3 mg/kg [146]. These data suggest that a minimal reinstatement of dopaminergic activity by apamin improved the apathetic behavior. With this toxic model, another team has shown that unilateral injection of 6-OHDA in the MFB leads to a decrease of the intake of sucrose 3%. At the first week post-injection, this apathetic behavior is found for the doses of 12 and 16 µg of 6-OHDA and only for the doses of 16 µg at 3 weeks post-injection [147]. Moreover, this apathetic effect of 6-OHDA was reversed by L-DOPA oral treatment (25 mg/kg) and partially by intraperitoneal injection of bupropion (10 mg/kg) but not by paroxetine (10 mg/kg) [148]. In operant procedure, the sucrose deliveries decrease drastically after 6-OHDA infusions in the SNc. This deficit is only reversed by the preferential agonist D3R PD-128907 but not by sumanirole or the selective agonist D1R SKF-38393 [124]. The results suggest a differential implication of the dopaminergic D1, D2 and D3 receptors in the motivational impairments induced by 6-OHDA SNc lesion, and the D3R is believed to be an important mediator of these beneficial effects.

Apathy-related behavior was also assessed in the MPTP mouse model. Even though this model presents anxious and/or depressive-like behaviors, data suggested a lack of apathetic behavior as assessed with SCT. Indeed, several studies are along these lines: 4 intraperitoneal injections of MPTP at a dose of 20 mg/kg (2 h interval) had no effect 7 and 30 days after MPTP injections [129,149].

In 12 month-old VMAT-2 deficient mice, saccharin (0.1%) preference was decreased compared to WT, without depressive behavior estimated with the forced swim test that might be related to dopaminergic dysfunction [131]. Moreover, heterozygous (HET) VMAT-2 mice present a decreased sucrose preference for 1% and 1.5% sucrose concentration solution whereas any anxiety-like behavior was evidenced with light/dark transition task [132].

Moreover, apathy-like behavior was found after an injection of 3 µL of AAV-6 expressing alpha-synuclein in the SNc, with 2% sucrose preference test during 48 h [141]. On the contrary, other studies demonstrated a lack of anhedonia in this viral model with a SCT realized with 2% sucrose during five days or with 3% sucrose during 1 h, respectively [114,142]. The apathetic behavior can also be assessed in single cages with two drinking spouts (saccharin at different doses or water, counterbalanced every day). After a nigral lesion by AAV-2 expressing human A53T alpha-synuclein under the synapsin promoter, no reduction of saccharin preference was found. The dose-responses test showed that lesioned animals were able to discriminate between different doses of saccharin after AAV-2 injections [150].

Whereas the sucrose preference was reduced, the underlying phenomenon remains unclear and can result from a decrease in the sensitivity to rewarding properties of the sucrose solution or be due to an extension of the lesion to other brain areas based on the Braak stages [5].

The different PD animal models showing neuropsychiatric disorders are summarized in Table 1.

### 3.2. Rodent Models of Cognitive Dysfunctions

Cognitive disturbances in animal models of PD have been more scarcely examined than, for example, motor deficits or treatment associated motor complications (dyskinesias). The fact that the subdivision of cognitive domains made in patients cannot be directly extrapolated in animals could have had an influence. At least two main factors have to be noted: first, animals do not process language in the same complex way as humans do, and thus this cognitive domain cannot be studied analogously. Second, the outcome of behavioral tasks that are used in animals typically require visual as well as motor performance, and therefore, results can be in part influenced by motor impairment. In any case, different animal models have shown some cognitive disturbances that cannot be solely attributed to motor impairment and are going to be summarized in the following sections and in Table 2.

#### 3.2.1. Attention

In regard to 6-OHDA rodent models of PD, some attentional deficits that are caused by fronto-striatal dysfunction in PD patients can also be analogously found in these animals. Thus, in the runway test, food deprived animals have to walk through a corridor that has food pellets in both sites. Interestingly, unilaterally lesioned rodents have shown a tendency towards ignoring the pellets in the corridor side contralateral to the lesion and to retrieve pellets from the side of the corridor ipsilateral to the lesion [151,152]. Analogous to this result, a study employing a lateralized choice task, where animals are required to respond to light stimulus presented in two adjacent ports to a central port, unilaterally lesioned rats within the MFB, displayed visuospatial neglect of the side opposite to the lesion [153]. These results may indicate specific deficits in attentional focusing for the lesioned hemisphere compared with the intact site.

On the other hand, rats with mild bilateral striatal dopaminergic depletion in the dorsolateral STR induced by 6-OHDA (60–80% loss of TH positive terminals) have shown no differences in comparison to control rats in the standard procedure of the 5-Choice Serial Reaction Time-Task (5-CSRTT). However, these rats displayed decreased accuracy and increased number of omissions when task conditions were manipulated to increase the attentional demand [154]. Similarly, another study has shown that after the development of the lesion, rats unilaterally lesioned in the MFB were not able to reach basal performance in a modified version of the 5-CSRTT. In addition, parkinsonian animals displayed impaired attention to stimuli both contralateral (20–30% accuracy) and ipsilateral to the lesion (60% accuracy), suggesting that deep unilateral dopamine depletion is sufficient to prompt a global selective and sustained attentional dysfunction. Interestingly, treatment with 10 mg/kg L-DOPA further impaired the performance, perhaps because it led to an increase of dopamine to a detrimental level in the intact hemisphere causing subsequent cortico-striatal network dysfunction [155].

Regarding other animal models, a recent study trying to obtain a rat model of PD and dopamine agonist induced impulsive behavior has shown that parkinsonian rats with partial bilateral nigrostriatal degeneration induced by human A53T α-synuclein over-expression within the SNc displayed a reduction of accuracy in the 5-Choice Serial Reaction time-task when compared to control rats or pre-parkinsonian status. In addition, chronic treatment with the dopaminergic agonist PPX worsened the attentional performance even more [156]. These results may reflect the clinical scenario where a loss of a proper dopaminergic tone caused by the disease within the prefrontal cortex or excessive input caused by dopaminergic treatment can both cause lack of an adequate behavioral control, thus impairing attentional performance.

#### 3.2.2. Memory Impairment and Learning Deficits

In relation to memory impairment, the Morris water maze (MWM) test has been extensively used in animal models of neurodegenerative disorders to study visuospatial memory. This task consists of a circular arena that is filled with opaque water, which usually hides a platform, and several modifications can be made to analyze different types of memory/learning processes. Thus, in the “cued version” of the task, the platform is always visible, animals are then trained by cue-guiding to find it. After training, animals are tested and the time to reach the platform is considered as a measurement of cued-learning (procedural-learning). On the other hand, with the “spatial” more standard version of the task, the platform is maintained under water and animals progressively learn to find it using different cues placed around the pool as spatial references. Thus, the time until reaching the platform during training is considered as a measurement of the learning-process. During testing, the platform is usually removed from the pool and the time spent by the animals in the place where the platform was previously located is taken as a measurement of the solidity of the acquired memory. Importantly, the performance in these two variations of the task are dependent on the dorsal STR and hippocampus, respectively [157]. Besides, some variations of the spatial version can also be used to study working short-term memory (such as, for example, when testing is performed just after the training period within the task).

Thus, rats with unilateral SNc lesions have shown that both cued learning and memory are impaired, but that long-term spatial memory remains intact [115]. Similar results were also found in mice with unilateral MPTP induced lesions in the SNc or systemic administration of MPTP [158]. In contrast to these results, animals with unilateral lesions in the MFB have shown decreased spatial memory performance in the open field hole board test that was reversed with L-DOPA [158]. In addition, a previous study using the MWM showed that rats with bilateral MFB lesion without remarkable changes in swimming performance with respect to controls displayed decreased learning and memory performance in both versions of the MWM [159]. Similarly, rats bilaterally lesioned with 6-OHDA and MPTP in the SNc displayed decreased performance in both working-memory and long-term memory variations of the MWM, with no significant differences in locomotor behavior [160].

Spatial memory performance has also been studied with other tasks. Thus, in the novel object recognition (NOR) test animals are trained for a certain period where they get familiarized to a given object. After training, animals are tested and given the tendency of rodents to be attracted by novelty, the time expend surrounding the new object or new location relative to the total time of object/location exploration during the test provides an idea of both novelty-seeking behavior and visuospatial memory, and changing the retention interval can be used to examine short-term or long-term memory [161]. Spatial mazes (such as T-Maze and Y-Maze) have also been classically used to analyze reference and working memory in rodents. There, alternative approaches are performed in order to analyze spontaneous and controlled alternation between the arms of the maze. Essentially, subjects are placed in the center and given their tendency to seek novelty, they tend to enter the arm visited less recently, and this behavior is measured.

Thus, rats bilaterally lesioned by intranigral injection of MPTP have shown impaired novelty discrimination during a short-delay test in the NOR, associated with a presence of cell loss in the hippocampal CA1 area [162]. Interestingly, rats with similar lesions also displayed working-memory dysfunction in the T-maze test. However, both working-memory and novelty discrimination as well as the CA1 cell loss were partly prevented by the chronic administration of the NMDA receptor antagonist MK-801, suggesting that these receptors could be involved in PD-related neuronal and behavioral dysfunction in PD [163]. Analogously, mice with systemic administration of MPTP showed progressive NOR impairment correlated with gradual impairment of long-term potentiation in the hippocampal CA1 area [164]. In contrast, mice with bilateral 6-OHDA lesions within the STR showed impaired long-term but not short-term recognition within the NOR test, which was not affected by the administration of the D2/D3 receptor agonist, PPX, but improved by L-DOPA or the D1 receptor agonist, SKF81297 [165].

Similarly, in a spatial object recognition test where changes are made in the location of objects rather than in objects themselves, mice injected with 6-OHDA in the STR were impaired in both the short- and the long-term object recognition, which seemed restored by the administration of a mGluR5 receptor selective antagonist. These results suggest that mGluR5 receptors could modulate dopamine release from the remaining nerve terminals, thus improving the associated acquisition and maintenance of spatial information [166]. In a version of the T maze where alternation of arms was controlled by food mediated reward, rats injected in the SNc with MPTP were shown to be able to learn and memorize the rewarded arm, but they failed to learn the new criteria when the rewarded arm changed [167]. Similarly, MPTP treated mice were not able to learn the criterion particularly when the duration of the delay between training and testing was variable, instead of fixed [168].

Other studies have also focused on analyzing the process of associative and emotional learning. For instance, rats bilaterally lesioned with MPTP in the SNc have shown impairments in the active avoidance task [169]. Within this task, subjects are trained to learn the presentation of a given cue with the subsequent occurrence of a negative event (such as a shock), which can be avoided by an action such as moving to a different place within the chamber. Thus, correct responses to the cue (measured as trials where escape is effectively performed) give an index of associative-learning.

Together these results suggest that toxin-based animal models of PD with unilateral nigrostriatal lesions usually suffer from fewer long-term memory deficits but might serve to study procedural learning deficits observed in PD patients, particularly at early stages. However, both unilateral lesions affecting SNc and VTA [158] and bilateral nigrostriatal denervation can simulate more advanced stages of the disease as well as simulate other clinical scenarios, such as impaired learning (including associative/emotional learning and procedural learning), impaired working memory and impaired long-term memory observed in patients.

Regarding transgenic rodent models of PD, although they represent an interesting tool to study different proteins and pathways implicated in PD pathophysiology, it has to be noted that, a large portion of them do not display clear motor dysfunction, and robust dopaminergic neuronal loss is seldom observed [170]. Several studies have been performed in transgenic models of PD as well, trying to analyze memory and learning. Thus, Thy1-aSyn transgenic mice (line 61), for example, display impairments in tests of novel object recognition (NOR), object–place recognition (spatial recognition), and operant reversal learning compared to age-mated wild-type littermates. Moreover, these deficits were noticeable before any marked dopaminergic cell loss in the SNc, but with remarkable deficits in the cholinergic cortical innervation [171]. Interestingly, this transgenic line also displays deficits in social cognition [172] Social cognition and NOR deficits in these mice were reversed by chronic nicotine administration [173]. Similarly, mice expressing human wild-type alpha-synuclein under the PDGF-β promoter also showed impaired MWM performance, which seems to be age-dependent and is reversed by immunotherapy or the mGLuR5 receptor antagonist MPEP (see [174] for review).

Other studies have also shown that mice over-expressing A30P mutated human alpha synuclein under the Thy1 promoter display age-dependent deficits in MWM, in 12 month-old but not in 4 month-old mice performing significantly worse than controls. Moreover, these older mice displayed widespread alpha-synucleinopathy in several brain regions critical for a proper cognitive control, such as the central nucleus of the amygdala [175]. Later, the same animal model was shown to have deficits in fear-conditioning as well [176]. Importantly, age-dependent development of cognitive impairment is a feature that is also present in other transgenic mice models of PD, such as the mice expressing human A53T alpha-synuclein under the mouse prion promoter, that develop deficits in the Y maze in 6 and 12 month old but not in 2 month old subjects [137]. Another study with the same model has also shown that these mice exhibited learning dysfunction in the MWM at 8 months of age, which is partly reversed by the treatment with the metal chelator Clioquinol [177]. Interestingly, cognitive impairment in this model of PD seems to be dependent on the presence of the protein Tau, since A53T mice lacking endogenous mouse tau expression do not display deficits in spatial learning and memory at 12 months [178].

The mouse model expressing Y39C mutant human alpha-synuclein under the Thy-1 promoter has also shown age-dependent accumulation of human alpha-synuclein oligomers in different areas of the brain, with older mice displaying cognitive impairment in the MWM and intracellular inclusions in neurons of different brain areas, particularly the cortex. However, and importantly, these mice did not show any significant loss of dopamine neurons in substantia nigra or dopamine nerve terminals in STR [179].

Since the effect of synuclein expression may vary depending on age, another study used transgenic mice conditionally expressing human wild-type alpha-synuclein in the midbrain and forebrain, and showed that these animals develop nigral and hippocampal neuropathology, including reduced neurogenesis and neurodegeneration in absence of fibrillary inclusions, which was in turn associated with memory impairment in the MWM and progressive motor decline [180].

Another study also demonstrated that MitoPark transgenic mice, which develop progressive neurodegeneration, loss of motor function and therapeutic response to L-DOPA, display reduced learning but intact long-term memory function when performing the Barnes maze test, as well as reduced NOR compared to age-matched non-transgenic littermates. Importantly, these features were present before the onset of motor impairment [181]. Similar learning and memory deficits were also observed in the Barnes maze in mice over-expressing the human truncated (1–120) form of alpha synuclein at 9 months of age. These mice already displayed deficits in NOR from 3 month and onwards when compared to WT littermates, which seems like it is associated with synuclein aggregates in different brain regions such as the hippocampus and cortex [182].

Regarding experiments in transgenic rat models of PD, an experiment using bacterial artificial chromosome (BAC) transgenic rats over-expressing human leucin-rich repeat kinase 2 protein (LRRK2) with G2019S and R1441C mutations has shown that these rats display age-dependent deficits in spontaneous alternation test of spatial short-term memory in spite of a lack of any dopaminergic cell death in the SNc [183].

Finally, another study has reported that rats injected in the VTA with viral vectors for over-expression of human wild-type alpha synuclein display impaired spatial learning and memory deficits in the MWM without any obvious spontaneous locomotor impairment and with the presence of alpha-synuclein positive inclusions in the hippocampus, neocortex, nucleus accumbens and anteromedial STR [184].

#### 3.2.3. Inhibitory Control

Animal models of PD represent an excellent tool to study inhibitory control deficits, and have been used in recent years in order to study PD-associated and dopaminergic treatment-associated changes in different impulsivity traits.

In relation to impulsive action, rats with bilateral parkinsonism induced by 6-OHDA injections in the SNc and VTA [185] or in the dorsolateral STR [156] do not display increased motor impulsivity in a variable delay-to-signal (VDS), a task that measures different components of impulsivity [186]. In contrast, rats with progressive parkinsonism induced by viral-vector mediated human A53T α-synuclein over-expression in the SNc have displayed increased motor and waiting impulsivity as measured by fixed consecutive number (FCN), differential reinforcement of low rate of responding (DRL) and 5-Choice Serial Reaction Time-Task (5CSRTT) [156,187]. Interestingly, the dopamine agonist PPX seemed to particularly increase impulsivity traits in those animals with higher pre-treatment impulsivity [156], suggesting that parkinsonism itself could act as a risk factor for treatment induced addictive behaviors in this model. Thus, although the impulsivity traits measured by different tasks in these studies may differ, these results may also indicate that the pattern and degree of striatal dopaminergic depletion could influence the increase of impulsive action, with α-synuclein models more prone to develop motor impulsivity after the dopaminergic lesion.

Regarding impulsive choice, rats with bilateral striatal denervation, induced by 6-OHDA, do not show any increment of delay-discounting [188] although another study with the same model of PD reported increased delay-discounting previously [189]. Although the lack of a control group within this last study makes it difficult to extract firm conclusions, the different reinforcers used in the two experiments could underlie this discrepancy (sucrose solution vs intracranial self-stimulation (ICSS) of amygdala, respectively). In this context, impulsive choice seems similar in rats lesioned in the dorsolateral STR with respect to control rats in a probabilistic discounting task that uses ICSS as a positive reinforcer [190]. Besides, and as happens with impulsive action, rats lesioned in the SNc and VTA or in dorsolateral STR with 6-OHDA do not display increased delay-intolerance with respect to control rats in the VDS task using food pellets as positive reinforcers [185,191].

Finally, although there are not studies specifically addressing compulsivity in PD patients, one study has reported that compulsive-like lever pressing as measured by post-signal training attenuation task is similar in rats lesioned by 6-OHDA injections in the SNc and VTA and controls, but that these rats, not the controls develop compulsive-like behavior after the administration of the dopamine agonist PPX [192]. Importantly, a more recent study has revealed that rats with intact nigrostriatal system but not rats with unilateral parkinsonism induced by 6-OHDA injections in the MFB developed of compulsive binge-eating. However, L-DOPA treatment is able to reinstate the capacity of developing compulsive binge-eating in these rats [193].

#### 3.2.4. Behavioral Flexibility

Deficits in behavioral flexibility were identified in rodent models of PD. Several tests were used to determinate these deficits as set-shifting task, reversal learning task or touch screen task in different protocols: operant procedure or mazes. Indeed, following bilateral 6-OHDA injections into the dorsomedial STR, lesioned rats displayed impairments during reversal learning but not in acquisition phases in a cross maze test [194]. However, in a touch screen task or a cross maze, unilateral or bilateral injections of 6-OHDA in the dorsolateral STR did not lead to deficits in behavioral flexibility although learning was impaired [195,196]. Another study has conducted an attentional set-shifting test (simple discrimination, reversal and extradimensional shift paradigms) in rats unilaterally lesioned in the MFB with 6-OHDA. The results have demonstrated an impairment in behavioral flexibility in both males and females, only in the simple discrimination with an increasing number of trials needed to complete the rule but not in reversal learning or extra-dimensional task with only an increase in time to complete these paradigms [197]. Moreover, a decrease of cognitive flexibility was demonstrated using a reversal learning task, but mice over-expressing human WT alpha-synuclein driven by the murine Thy-1 promoter (line 61) displayed impairments only in the reversed rule, not in the acquisition phase [171]. Another study used a genetic model of PD with a depletion of dopamine synthesis under control of Cre manipulation (conditional tyrosine hydroxylase knock-out mouse) without dopaminergic neuron loss [198,199]. This model resulted in a slight impairment in behavioral flexibility task (U-maze), similar to that of mice injected with 6-OHDA in the dorsal STR [199]. However, these two groups of mice were able to learn the rule of the task.

**Table 2 biomedicines-09-00684-t002:** Overview of cognitive deficits in rodent models of PD. +/− = presence or absence depending on studies, + + = presence, n.r. = non reported.

Model	Flexibility Deficits	Inhibitory Control Deficits	Attention Deficits	Learning Deficits	Short-Term Memory Impairment	Long-Term Memory Impairment	Ref
6-OHDA (Bilateral, STR)	+/−	Discrepancies	+ + (if attentional load increased)	n.r.	+ +	+ +	[154,156,165,188,189,190,191]
6-OHDA (Bilateral, SNc)	n.r	n.r.	n.r.	n.r.	+ +	+ +	[160]
6-OHDA (Bilateral, SNc + VTA)	n.r.	Intact	n.r.	n.r.	n.r.	n.r.	[185]
6-OHDA (Bilateral, MFB)	n.r.	n.r.	n.r.	n.r.	+ +	+ +	[151,159]
6-OHDA (Unilateral, SNc)	n.r.	n.r.	n.r.	+ +	+ +	Intact	[115]
6-OHDA (Unilateral, MFB)	+ +	Reduced tendency to binge-eating	+ + (particularly ipsilateral to the lesion)	n.r.	n.r.	+ +	[153,155,158,193]
MPTP (bilateral, systemic administration)	n.r.	n.r.	n.r.	+ + (only for learning new rules)	+ +	Intact	[168]
MPTP (unilateral, SNc)	n.r.	n.r.	n.r.	+ +	+ +	Intact	[158]
MPTP (bilateral, SNc)	n.r.	n.r.	n.r.	+ +	+ +	+ +	[158,160,167,169]
hA53T alpha-synuclein mice (mouse prion promoter)	n.r.	n.r.	n.r.	+ + (older mice)	n.r.	+ + (older mice)	[137,177,178]
TH Knock-Out mice (Cre recombinase)	+ +	n.r.	n.r.	n.r.	n.r.	n.r.	[198,199]
WT alpha-synuclein mice (mouse Thy-1 promoter) line 61	+ +	n.r.	n.r.	+ +	+ +	+ +	[171,173]
WT alpha-synuclein mice (PDGF-β promoter)	n.r.	n.r.	n.r.	n.r.	n.r.	+ +	[174]
A30P alpha-synuclein mice (mouse Thy-1 promoter)	n.r.	n.r.	n.r.	+ + (older mice)	n.r.	+ + (older mice)	[175,176]
Y39C alpha-synuclein mice (mouse Thy-1 promoter)	n.r.	n.r.	n.r.	n.r.	n.r.	+ + (older mice)	[179]
WT alpha-synuclein mice (conditional expression in the midbrain and forebrain)	n.r.	n.r.	n.r.	n.r.	n.r.	+ +	[180]
Truncated 1-120 alpha-synuclein mice	n.r.	n.r.	n.r.	+ +	n.r.	+ +	[182]
MitoPark mice	n.r.	n.r.	n.r.	+ +	n.r.	+ +	[181]
BAC LRRK2 transgenic rats (G2019S and R1441C mutations)	n.r.	n.r.	n.r.	n.r.	+ +	n.r.	[183]
BAC WT alpha-synuclein transgenic rats	n.r.	n.r.	n.r.	+ +	n.r.	n.r.	[140]
AAV alpha-syn (VTA)	n.r.	n.r.	n.r.	+ +	n.r.	+ +	[184]
AAV A53T alpha-syn (SNc)	n.r.	+ +	+ +	n.r.	n.r.	n.r.	[156,187]

#### 3.2.5. Dementia

Because dementia is like complex NMS occurring when the neuropsychological profile of patients is markedly impaired in several cognitive domains, the animal models presenting several neuropsychiatric and cognitive impairments of considerable severity may be considered as a proxy for dementia. It is however challenging to really model dementia in animals and the magnitude of neuropsychiatric and cognitive impairments observed in animal models remain very far from dementia occurring in advanced stages of PD.

## 4. Clinical Relevance and Future Directions

The wide spectrum of neuropsychiatric and cognitive NMS of PD, and their important contribution to the overall severity of the disease and decreased quality of life is of significant concern. If some of these NMS are partially responsive to dopamine replacement therapy, others remain untreated due to the lack of therapeutic options. Toxin-based models have been useful to highlight the significant contribution of dopamine to some extent and other monoamines in the pathophysiology of these non-motor features of PD, while gene-based models have offered the opportunity to explore the contribution of cortical and subcortical alpha-synuclein pathology in the etiology of these disorders. Even if rodent models of PD only partially recapitulate the spectrum of neuropsychiatric and cognitive non-motor deficits found in the human disease, they remain valuable to understand their pathophysiology and for the preclinical validation of therapeutic strategies against th pervasive and incapacitating features of PD.

## Figures and Tables

**Figure 1 biomedicines-09-00684-f001:**
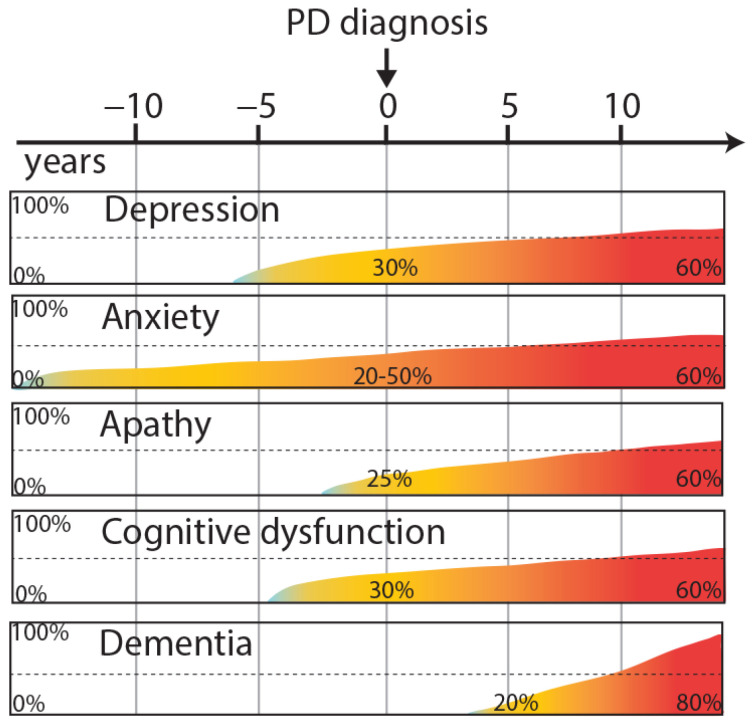
Overview neuropsychiatric and cognitive symptoms in PD, illustrating their approximate progression and temporal relationships with the onset of motor symptoms.

**Table 1 biomedicines-09-00684-t001:** Overview of neuropsychiatric deficits in rodent models of PD. ↓ = decrease, − − = absence, +/− = presence or absence depending on studies, + + = presence, n.r. = non reported, mo = months.

Model	Pathophysology	Depressive Symptoms	Anxiety	Apathy	Ref
Paraquat	↓ motor performance	+ +	+ +	n.r.	[113]
6-OHDA (Bilateral, STR)	↓ motor performance ↓ number of TH neurons (SNc) ↓ dopamine level in STR	+ +	+ +	+ +	[115,119,146]
6-OHDA (Bilateral, SNc)	↓ TH + neurons in SNc	+ +	+ +	+ +	[120,144,145]
6-OHDA (Unilateral, MFB)	96% loss of TH (STR); 87% loss of TH neurons (SNc)	+ +	+ +	+ +	[117,118,147]
MPTP (bilateral, systemic administration)	Motor impairment & ↓ TH + neurons in SNc	+ +	+ +	− −	[126,127,128,129,149]
VMAT-2 deficient mice	↓ striatal dopamine level↓ dopaminergic neurons of SNc alpha-synuclein agregation	+ + (12–15 mo) − − (12 mo)	− − (3–5 mo) + + (6 mo)	+ + (12 mo)	[130,131,132]
Parkin deficient mice (deletion of exon 3)	No significant motor impairments (6–21 months)	n.r.	+/− (6 and 15 mo)	n.r.	[133,134,135]
hA53T alpha-synuclein mice (mouse prion promoter)	Inclusions in several brain areas altered neuronal morphology, ↓ TH+ in SNc at 12 months	+ + (6 mo)	+ + (3 and 6 mo) − − (12 mo)	n.r.	[136,137,138,139,140]
BAC WT alpha-synuclein transgenic rats	Age-dependent accumulation of alpha-synuclein aggregates (mainly striatum)	n.r.	+ +	n.r.	[140]
AAV alpha-syn (SNc)	Progressive motor impairment, alpha-synuclein agregation, ↓ DA striatum and ↓ TH+ neurons in SNc	+/−	− −	− −	[114,141,142,150]

## Data Availability

Not applicable.

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
