# Peer review of "Neuropsychiatric and Cognitive Deficits in Parkinson’s Disease and Their Modeling in Rodents"

_biomedicines, 2021, doi:10.3390/biomedicines9060684_

Round 1
Reviewer 1 Report
The manuscript titled ‘Neuropsychiatric deficits in Parkinson’s Disease and their modelling in rodents’ by Decourt et al., is a comprehensive review of non-motor symptoms (NMS) in Parkinson’s disease. The manuscript is well structured- describing neuropsychiatric NMS in detail, including depression, anxiety, apathy, and cognitive dysfunction. They further discuss important findings from animal studies and provide insights into their shortcomings.
This informative review stresses how critical it is to detect NMS early as it can be used to predict disease outcomes and provide early interventions. The manuscript is well organized and provides an in-depth analysis of each subtopic.
Author Response
We thank reviewer 1 for his/her kind comments on the quality of the article.
Reviewer 2 Report
The main aim of the present review was to provide an overview about the neurological non—motor symptoms associated to Parkinson’s disease (PD). Overall the manuscript is very well-written and provides an in depth description of the presence of these symptoms in PD patients and the ability of toxin- and genetic-induced animal models to recapitulate these manifestations.
Nevertheless, the title of the manuscript and the nomenclature used to describe the symptoms addressed in this review is somewhat confusing. The term neuropsychiatric deficits is suited for depression and other mood disorders, but does not seem adequate for the cognitive manifestations, such as dementia. Perhaps the use of this two dimensions “mood and cognition” would be better to describe the symptomatology described in the review. This type of confusion is also present throughout the paper, for instance in the legend of table 1.
Moreover, there are other issues regarding the structure of the manuscript, such as:
- Sections numbering
- The existence of abbreviations that are not described in the text, such as HC (page 6)
- The definition of abbreviations that had already been used in the text, such as NBM
- The vertical orientation and overall structure of table 1 impedes its readability
Author Response
We thank the reviewer for his/her thoughtful comments on the manuscript. We agree with the reviewer that these two dimensions where not properly distinguished in some of the paragraphs and sections of the article. Therefore, we have changed the text accordingly to better distinguish between Mood/neuropsychiatric deficits on one hand, and cognitive deficits on the other hand. The title of the article has also been updated.
Moreover, there are other issues regarding the structure of the manuscript, such as:
- Sections numbering
- The existence of abbreviations that are not described in the text, such as HC (page 6- The definition of abbreviations that had already been used in the text, such as NBM
All the abbreviations are now described at their first occurrence in the text and section numbering has been updated.
- The vertical orientation and overall structure of table 1 impedes its readability
To improve the readability, and offer a better distinction between mood/neuropsychiatric deficits and cognitive dysfunctions, there are now two tables in the revised version: one for mood/neuropsychiatric deficits (Table 1) and one for cognitive deficits (Table 2)
Reviewer 3 Report
This review summarized the non-motor symptoms in Parkinson's disease patients and rodent pre-clinic models. The authors focused on mood and neuropsychiatric disorders, mainly depression, anxiety, apathy, cognitive dysfunction, and dementia. It provided a comprehensive summary of the neuropsychiatric deficits in PD and PD models with an in-depth analysis of the possible molecular mechanism behind the symptoms. The manuscript was well written with sufficient background information on the behavioral tests for each psychiatric deficit in PD models. The included table offered a concise and clear summary of rodent models of PD in the literature.
Only a few comments for improvement:
- The authors heavily relied on one particular paper [DoPaMiP study] in the PD mood disorders section. It has been cited over five times in the two paragraphs. Including more PD studies might make it more balanced and comprehensive.
- In Figure 1, the authors composed a cartoon overview of the progression of neuropsychiatric symptoms in PD. However, this progression chart was primarily based on estimation with very limited patients with 2-3 time points in different studies. The depicted linear progressions in most of the symptoms were not evidence-based.
- Disorganized section 2. There were multiple 2.1, 2.2, and 2.3 in the manuscript.
- Citations need for the statement "SSRI in depressive PD patients are not always optimum" in 2.1 mood disorders.
- More discussion of the "retrieval failure hypothesis" might help readers understand the conflicting results in declarative episodic memory deficits in PD literature.
- Same as point #5, the authors could provide a more explicit statement on the disagreement of CA1 vs. CA2-3 as the main contribution for episodic recollection in the literature.
- The authors should use full names before introducing the abbreviations, such as HC in 2.2 Declarative emotional memory and reward-learning, ICBs in 2.4 Inhibitory control, and STR in 2.5 Behavioral flexibility.
- Mixed citation format in Willner et al., 1987 and Tedford et al.,2015.
- Incomplete statement in "a previous study using the MWM showed that rats with bilateral MFB." MFB lesion?
- Please provide a legend for the symbols used in Table 1.
Author Response
Only a few comments for improvement:
- The authors heavily relied on one particular paper [DoPaMiP study] in the PD mood disorders section. It has been cited over five times in the two paragraphs. Including more PD studies might make it more balanced and comprehensive.
We thank the reviewer for his/her kind comments on the quality of the manuscript. We have now included more references on this topic, including the ONSET PD study. (Pont-Sunyer et al., Mov Disord 2015).
- In Figure 1, the authors composed a cartoon overview of the progression of neuropsychiatric symptoms in PD. However, this progression chart was primarily based on estimation with very limited patients with 2-3 time points in different studies. The depicted linear progressions in most of the symptoms were not evidence-based.
We have updated the figure legend to highlight that this overview is approximate and based on a limited number of time-points available from the literature.
- Disorganized section 2. There were multiple 2.1, 2.2, and 2.3 in the manuscript.
The organization of section 2 has been modified accordingly
- Citations need for the statement "SSRI in depressive PD patients are not always optimum" in 2.1 mood disorders.
We have included a reference addressing this statement: Liu J, Dong J, Wang L, Su Y, Yan P, Sun S. Comparative Efficacy and Acceptability of Antidepressants in Parkinson’s Disease: A Network Meta-Analysis. PLoS ONE 2013;8:e76651. https://doi.org/10.1371/journal.pone.0076651.
- More discussion of the "retrieval failure hypothesis" might help readers understand the conflicting results in declarative episodic memory deficits in PD literature.
Following the reviewers’ recommendations, we have updated the text. In this regard, although the retrieval failure hypothesis has been considered as the main factor behind the discrepancies between studies addressing memory deficits in early PD, according to the latest knowledge, these differences in the published outcomes could be directly explained by deficiencies in the learning process that does occur in PD patients from the very beginning, and that have been classically excluded. In order to make this newer scenario more comprehensive for the audience, the text has been extended and these two facts have been further explained. Now, the paragraph reads as follows:
« Memory seems to be the most commonly impaired cognitive domain at baseline in PD [63]. However, memory deficits in PD patients without dementia are substantially improved after cueing [64]. Thus, this may indicate that these subjects have intact ability to store the information in long term memory, but that there is an impairment in the process of accessing this stored information. This failure of evocation rather than storage is known as “retrieval failure hypothesis”. However, one study specifically assessing learning abilities in non-demented PD subjects has demonstrated that these subjects indeed display impaired ability to acquire new information [65]. Therefore, in addition to a role for executive dysfunction and decreased attentional performance [60], declarative memory dysfunction reported in the first stages of the disease could be driven by difficulties in the processing of explicit learning. In this regard, the lack of controlling for initial learning in the majority of studies addressing long-term memory performance in non-demented PD patients could directly affect the observed outcomes. This fact should be further addressed in future studies. »
- Same as point #5, the authors could provide a more explicit statement on the disagreement of CA1 vs. CA2-3 as the main contribution for episodic recollection in the literature.
We agree with the reviewer’s statement regarding this discrepancy. Since there is currently no consensus regarding the respective contribution of CA1 vs CA2-3 on episodic recollection deficits, we have mentioned in this revised version that this topic deserves further investigation.
- The authors should use full names before introducing the abbreviations, such as HC in 2.2 Declarative emotional memory and reward-learning, ICBs in 2.4 Inhibitory control, and STR in 2.5 Behavioral flexibility.
All the abbreviations within the text and the text itself have been reviewed again, and these issues have been corrected accordingly.
- Mixed citation format in Willner et al., 1987 and Tedford et al.,2015.
These referencing errors have been corrected within the text.
- Incomplete statement in "a previous study using the MWM showed that rats with bilateral MFB." MFB lesion?
The word « lesion » has been included within the text
- Please provide a legend for the symbols used in Table 1.
A legend has been added and the table has been split in two to answer the comment from reviewer #1